# Human exome and mouse embryonic expression data implicate *ZFHX3*, *TRPS1*, and *CHD7* in human esophageal atresia

Rong Zhang[1], Jan Gehlen[2], Amit Kawalia[3], Maria-Theodora Melissari[4], Tikam Chand Dakal[5], Athira M. Menon[5], Julia Höfele[6], Korbinian Riedhammer[6,7], Lea Waffenschmidt[1], Julia Fabian[1], Katinka Breuer[1], Jeshurun Kalanithy[1], Alina Christine Hilger[8], Amit Sharma[9,10], Alice Hölscher[11], Thomas M. Boemers[11], Markus Pauly[12], Andreas Leutner[13], Jörg Fuchs[14], Guido Seitz[15], Barbara M. Ludwikowski[16], Barbara Gomez[16], Jochen Hubertus[17], Andreas Heydweiller[18], Ralf Kurz[18], Johannes Leonhardt[19], Ferdinand Kosch[20], Stefan Holland-Cunz[21], Oliver Münsterer[22], Beno Ure[23], Eberhard Schmiedeke[24], Jörg Neser[25], Petra Degenhardt[26], Stefanie Märzheuser[27], Katharina Kleine[28], Mattias Schäfer[29], Nicole Spychalski[29], Oliver J. Deffaa[30], Jan-Hendrik Gosemann[30], Martin Lacher[30], Stefanie Heilmann-Heimbach[1,31], Nadine Zwink[32], Ekkehart Jenetzky[32,33], Michael Ludwig[34], Phillip Grote[4], Johannes Schumacher[1,2], Holger Thiele[3], Heiko Reutter[1,35]*

**1** Institute of Human Genetics, Medical Faculty of Bonn, University of Bonn, Bonn, Germany, **2** Institute of Human Genetics, University Hospital of Marburg, Marburg, Germany, **3** Cologne Center for Genomics, University of Cologne, Cologne, Germany, **4** Institute of Cardiovascular Regeneration, Center for Molecular Medicine, University of Frankfurt, Frankfurt am Main, Germany, **5** Department of Biotechnology, Mohanlal Sukhadia University Udaipur, Rajasthan, India, **6** Institute of Human Genetics, Klinikum Rechts der Isar, Technical University of Munich, School of Medicine, Munich, Germany, **7** Department of Nephrology, Klinikum Rechts der Isar, Technical University of Munich, School of Medicine, Munich, Germany, **8** Department of Pediatrics, Clinic for Pediatrics, University Hospital Bonn, Bonn, Germany, **9** Department of Neurology, University Hospital Bonn, Bonn, Germany, **10** Department of Ophthalmology, University Hospital Bonn, Bonn, Germany, **11** Department of Pediatric Surgery and Urology, University Hospital Cologne, Cologne, Germany, **12** Department of Pediatric Surgery, Asklepios Children's Hospital St. Augustin, St. Augustin, Germany, **13** Department of Pediatric Surgery, Medical Center Dortmund, Dortmund, Germany, **14** Department of Pediatric Surgery Children's Hospital, University of Tübingen, Tübingen, Germany, **15** Department of Pediatric Surgery, University of Marburg, Marburg, Germany, **16** Department of Pediatric Surgery and Pediatric Urology, Medical Center for Children and Adolescents AUF DER BULT, Hannover, Germany, **17** Department of Pediatric Surgery, Dr. von Hauner Children's Hospital, Ludwig-Maximilians-University, Munich, Germany, **18** Department of Pediatric Surgery, University Hospital Bonn, Bonn, Germany, **19** Department of Pediatric Surgery, Children's Hospital Braunschweig, Braunschweig, Germany, **20** Department of Pediatric Surgery, Städtisches Klinikum Karlsruhe, Karlsruhe, Germany, **21** Department of Pediatric Surgery, University Children's Hospital Basel (UKBB), Basel, Switzerland, **22** Department of Pediatric Surgery, University Medicine Mainz, Mainz, Germany, **23** Center of Pediatric Surgery Hannover, Hannover Medical School, Hannover, Germany, **24** Clinic for Paediatric Surgery and Paediatric Urology, Klinikum Bremen-Mitte, Bremen, Germany, **25** Department of Pediatric Surgery, General Hospital, Chemnitz, Germany, **26** Department of Pediatric Surgery, Ernst von Bergmann Hospital, Potsdam, Germany, **27** Department of Pediatric Surgery, Campus Virchow Clinic, Charité University Hospital Berlin, Berlin, Germany, **28** Department of Pediatric Surgery, Evangelisches Krankenhaus Oberhausen, Germany, **29** Department of Pediatric Surgery and Urology, Cnopf'sche Kinderklinik, Nürnberg, Germany, **30** Department of Pediatric Surgery, University of Leipzig, Leipzig, Germany, **31** Department of Genomics, Life &Brain Center, University of Bonn, Bonn, Germany, **32** Department of Child and Adolescent Psychiatry and Psychotherapy, Johannes-Gutenberg University, Mainz, Germany, **33** Institute of Integrative Medicine, Witten/Herdecke University, Herdecke, Germany, **34** Department of Clinical Chemistry and Clinical Pharmacology, University of Bonn, Bonn, Germany, **35** Section of Neonatology and Pediatric Intensive Care, Clinic for Pediatrics, University Hospital Bonn, Bonn, Germany

☯ These authors contributed equally to this work.

\* reutter@uni-bonn.de



**Data Availability Statement:** All relevant data are within the manuscript and its Supporting Information files.

**Funding:** F.K. was supported by a stipend of the University of Bonn, BONFOR (O-149.0115.1; https://www.medfak.uni-bonn.de/de/forschung/foerderung/interne-foerderung/bonfor). H.R., J.S., M.L., and P.G. are supported by the German Research Foundation (Deutsche Forschungsgemeinschaft, DFG) (BE 3910/6-1, RE 1723/4-1, Exc147-2; https://www.dfg.de/). H.R., J.S., H.T, and E.J. are further supported by a grant of the Else-Kröner-Fresenius-Stiftung (EKFS, 2014_A14; https://www.ekfs.de/). The Exome analysis was performed on CHEOPS, a high performance computer cluster of the regional data center (RRZK) of the University of Cologne, funded by the DFG (215828658). The transcriptome analysis were performed on the de.NBI cloud, a national infrastructure supported by the German Federal Ministry of Education and Research (FKZ 031A532-0331A540 and 031L0101-0310108). The funders had no role in study design, data collection and analysis, decision to publish, or preparation of the manuscript.

**Competing interests:** The authors have declared that no competing interests exist.

# Abstract

## Introduction

Esophageal atresia with or without tracheoesophageal fistula (EA/TEF) occurs approximately 1 in 3.500 live births representing the most common malformation of the upper digestive tract. Only half a century ago, EA/TEF was fatal among affected newborns suggesting that the steady birth prevalence might in parts be due to mutational *de novo* events in genes involved in foregut development.

## Methods

To identify mutational *de novo* events in EA/TEF patients, we surveyed the exome of 30 case-parent trios. Identified and confirmed *de novo* variants were prioritized using *in silico* prediction tools. To investigate the embryonic role of genes harboring prioritized *de novo* variants we performed targeted analysis of mouse transcriptome data of esophageal tissue obtained at the embryonic day (E) E8.5, E12.5, and postnatal.

## Results

In total we prioritized 14 novel *de novo* variants in 14 different genes (*APOL2*, *EEF1D*, *CHD7*, *FANCB*, *GGT6*, *KIAA0556*, *NFX1*, *NPR2*, *PIGC*, *SLC5A2*, *TANC2*, *TRPS1*, *UBA3*, and *ZFHX3*) and eight rare *de novo* variants in eight additional genes (*CELSR1*, *CLP1*, *GPR133*, *HPS3*, *MTA3*, *PLEC*, *STAB1*, and *PPIP5K2*). Through personal communication during the project, we identified an additional EA/TEF case-parent trio with a rare *de novo* variant in *ZFHX3*. *In silico* prediction analysis of the identified variants and comparative analysis of mouse transcriptome data of esophageal tissue obtained at E8.5, E12.5, and postnatal prioritized *CHD7*, *TRPS1*, and *ZFHX3* as EA/TEF candidate genes. Re-sequencing of *ZFHX3* in additional 192 EA/TEF patients did not identify further putative EA/TEF-associated variants.

## Conclusion

Our study suggests that rare mutational *de novo* events in genes involved in foregut development contribute to the development of EA/TEF.

## Introduction

Esophageal atresia with or without tracheoesophageal fistula (EA/TEF) occur approximately 1 in 3.000 to 3.500 live births representing the most common malformation of the upper digestive tract [1; 2; 3; 4]. According to the "European network of population-based registries for the epidemiological surveillance of congenital anomalies (EUROCAT) EA/TEF account for 1% of all birth defects in Europe every year (https://eu-rd-platform.jrc.ec.europa.eu/eurocat). Hence, with 5.075 million babies born in the EU in 2017, 1.237 babies have been born with EA/TEF.

In about 40–50% of cases, EA/TEF occurs within the context of additional anomalies mostly belonging to the VATER/VACTERL association (OMIM #192350) spectrum. This acronym

refers to the rare, nonrandom co-occurrence of the following component features (CFs): vertebral defects (V), anorectal malformations (A), cardiac defects (C), tracheoesophageal fistula with or without esophageal atresia (TE), renal malformations (R), and limb defects (L) [5]. Only half a century ago, EA/TEF was fatal among affected newborns suggesting that the steady birth prevalence might in parts be due to mutational *de novo* events in genes involved in foregut development. Support for this hypothesis comes from early reports of chromosomal *de novo* aberrations present in 6–10% of syndromic EA/TEF cases [6]. Furthermore, using copy number variation (CNV) analysis in 375 EA/TEF patients we identified eight rare CNVs in six patients, all of which occurred *de novo*, including one CNV previously associated with EA/TEF [7]. Hence, 1.55% of isolated EA/TEF patients and 1.62% of patients with additional congenital anomalies carried *de novo* CNVs. Moreover, several monogenic EA/TEF associated syndromes are caused by smaller *de novo* changes comprising single nucleotides or small indels e.g. *N-MYC* in Feingold syndrome (OMIM #164280), *GLI2* in Pallister-Hall syndrome (OMIM #146510), *CHD7* in CHARGE syndrome (OMIM #214800), and *SOX2* in AEG syndrome (OMIM #206900) [8; 9; 10; 11].

To further explore the involvement of small genetic *de novo* events in the etiology of EA/TEF, we profiled 30 case-parents trios using exome sequencing (ES). Prior to ES chromosomal microarray analysis was negative in all cases [7; 12]. All confirmed *de novo* variants were prioritized using *in silico* prediction tools. To investigate the embryonic role of genes harboring prioritized *de novo* variants we performed targeted analysis of mouse transcriptome data of esophageal tissue obtained at embryonic day (E) E8.5, E12.5, and postnatal.

## Materials and methods

### Patients and DNA isolation

In 2011, the authors JS and HR founded the scientific network "great" (genetic risk for esophageal atresia; www.great-konsortium.de). The "great network" was founded in order to initiate a nationwide investigation into the genetic causes of EA/TEF. Prior to the commencement of recruitment, the network partners generated a unique standardized case report form (CRF). The CRF comprises an epidemiological questionnaire and a clinical assessment battery. The epidemiological questionnaire is based on: (i) the National Birth Defect Prevention Study questionnaire of the U.S. Centers of Disease Control and Prevention (www.nbdpn.org); and (ii) the questionnaire of the European Surveillance of Congenital Malformations (EUROCAT) network (www.eurocat-network.eu). The clinical assessment battery comprises the classification system of the EA/TEF phenotype according to Gross (1953), and the ICD10 coding with the British Pediatric Association one digit extension (www.eurocat-network.eu/content/EUROCAT-Guide-1.3.pdf) for classification of additional congenital anomalies. The great cohort is being recruited with the support of pediatric surgical departments across Germany, and the German self-help organization for patients and families with EA/TEF (KEKS e.V.; www.keks.org). KEKS e.V. is the largest self-help organization for EA/TEF families in Europe, and supports both the ongoing great investigations and the present proposal.

The here described study fulfilled the requirement of the Declaration of Helsinki and ethical approval was obtained from the local ethic committee of the Medical Faculty of Bonn (Lfd. Nr. 073/12). Every participating family provided written informed consent. The 30 here reported case-parent trios as well as the EA/TEF cohort for resequencing of *ZFHX3*, were recruited through the efforts of the scientific network "great". In 14 of the 30 case-parent trios, EA/TEF occurred isolated/nonsyndromic. In the remaining case-parent trios EA/TEF co-occurred with additional phenotypic features (syndromic cases) mostly belonging to the VATER/VACTERL spectrum (S1 Table). From each case-parent trio, EDTA blood samples were obtained.

Genomic DNA was isolated using the Chemagic DNA Blood Kit special (Chemagen, Baesweiler, Germany). Through personal communication we identified another patient with EA/TEF as part of his VATER/VACTERL association (patient 750_501).

## Exome Sequencing (ES) and data analysis

Exome capture was performed using the NimbleGen SeqCap EZ Human Exome Library v2.0 enrichment kit and sequenced with an Illumina paired end 2x100 bp sequencing (protocol v1.2). Primary data was filtered according to signal purity by the Illumina Realtime Analysis (RTA) software v1.8. Subsequently, reads were mapped to the human genome reference build hg19 using the bwa-aln [13] alignment algorithm. GATK v1.6 [14] was used to mark duplicated reads, for local realignment around short insertions and deletions, to recalibrate the base quality scores and to call SNVs (incorporating variants quality score recalibration) and short indels [15]. Scripts developed in-house at the Cologne Center for Genomics (unpublished) were used to incorporate allele frequencies reported by the ESP6500 database [Exome Variant Server, NHLBI GO Exome Sequencing Project (ESP), Seattle, WA (URL: http://evs.gs. washington.edu/EVS/)] and to detect changes in the protein structure. Acceptor and donor splice site mutations were analyzed with a Maximum Entropy model [16]. *De novo* variant calling was performed with the program DeNovoGear (v.0.5.1) [17] The Varbank GUI (unpublished, https://varbank.ccg.uni-koeln.de) was used to filter for high quality (coverage>15; quality>25), rare (MAF<0.005), *de novo* (posterior probability of a *de novo* mutation = PP_DNM>0.5) variants predicted to alter protein structure or splicing. We also filtered against an in-house database containing all variants from 511 exomes from epilepsy patients to exclude pipeline-related artefacts (MAF<0.004). Variants with MAF<0.004 that have been described to occur homozygous in gnomAD were also excluded. Finally, we further excluded all variants with a MAF≥0.0003 since the EA/TEF birth prevalence has been reported to be 1 in 3.500 live births (frequency of ≈ 0.0003). Hence, (full penetrant) monoallelic variants with a MAF≥0.0003 cannot account for the occurrence of EA/TEF.

## Variant validation and classification

Variants identified by ES were validated by using polymerase chain reaction (PCR). Automated sequence analysis was carried out using standard procedures. In brief, primers were directed to all variants observed and the resultant PCR products were subjected to direct automated BigDye Terminator sequencing (3130XL Genetic Analyzer, AppliedBiosystems, Foster-City, California, USA). Both strands from each amplicon were sequenced for the presence of these variants in the respective case-parent trio. In order to further prioritize the identified and confirmed *de novo* variants, we analyzed them using ten different *in silico* prediction tools which are encountered in dbNSFP v3.0 (https://sites.google.com/site/jpopgen/dbNSFP): SIFT, LRT, MutationTaster, Mutation Assessor, FATHMM, PROVEAN, MetaSVM, MetaLR, fathmm-MKL coding and CADD [18; 19] (details about these prediction tools are given as supporting information S1 Data).

## Re-sequencing of *ZFHX3* in EA/TEF patients

All three human *ZFHX3* protein coding transcripts (ENST00000641206.2, ENST00000268489.10, and ENST00000397992.5) listed in 'ensembl database' (www.ensembl. org/ Ensembl Release 98 (September 2019)) were sequenced in 192 unrelated EA/TEF patients. PCR-amplified DNA products (primer sequences available upon request) were subjected to sequencing using a 3130XL Genetic Analyzer (Applied Biosystems, Foster City, USA).

## Structural modeling and in-silico analysis of ZFHX3 protein variants

The secondary structure prediction of human OCTs protein sequences was done using PSIPRED in I-Tasser. Three-dimensional protein structural models for ZFHX3 were built using SWISS-MODEL (https://swissmodel.expasy.org/). Since, Swiss model cannot handle large protein sequence, for the prediction of ZFHX3 *de novo* changes p.Pro534Arg and p. Ala2126Val we trimmed the sequence of 60 amino acids upstream and 20 downstream of the mutated site. The sequence was subjected to swiss-model based modeling. The structural comparison between wild-type and mutant variant was done in Chimera after superimposing the structure of mutant onto the wild structure using SuperPose using default parameters (super-pose.wishartlab.com).

## RNA isolation and mRNA library preparation of mouse embryonic esophageal tissue

All animals used in this study were anesthetized by Isoflurane and killed by cervical dislocation. The animals that were used in this study are documented and their usage reported to the local authorities Regierungspräsidium Darmstadt). Embryos from pregnant females of the C57Bl6J strain were harvested at embryonic days (E) E8.5, E12.5, and postnatal. The embryos of the E8.5 litter were determined to be of the developmental Theiler stage 13 (TS13) and the E12.5 embryos TS21. From E8.5 embryos, the pharyngeal pouch containing endoderm and adjacent mesoderm tissue was surgically isolated and transferred into QIAzol®. Multiple embryos were pooled for each embryonal timepoint. For the E8.5 stage we pooled biopsies from 5 embryos to prepare the RNA and for the E12.5 and neonates we pooled two each for RNA preparation. From E12.5 and postnatal embryos, the distinct structures of the esophagus and the trachea was surgically isolated, combined and transferred into QIAzol®. RNA was isolated from these tissues with the RNEasy Mini Plus Kit (Qiagen) according to the manufacture's protocols. The transcriptome profile was assessed by RNA-Sequencing with the 3'-mRNASeq Library Preparation Kit from Lexogen, (Lexogen, Vienna, Austria). This protocol generates for each transcript only one single-end strand specific fragment for sequencing at the 3'-end of poly(A)-RNA. Libraries were quality checked on a TapeStation2200 (Agilent, Santa Clara, USA). The sequencing was performed on a HiSeq 2500 (Illumina, San Diego, USA) with two technical replicates of sample.http://www.bioinformatics.babraham.ac.uk/projects/

## Transcriptome analysis

https://www.ncbi.nlm.nih.gov/pmc/articles/PMC2654802/After demultiplexing with bcl2fastq (Illumina, San Diego, USA), FastQC v0.11.8 (http://www.bioinformatics.babraham.ac.uk/projects/) was used for quality control of FASTQ files. Read alignment was performed using STAR_2.6.1d [20] against the primary assembly of murine genome reference build GRCm38 according to the manufacturer's analysis protocol. Read counting was also performed with STAR ("quantMode GeneCounts") using the Ensembl gene annotation (Release 97). Quality metrics were gathered with multiQC [21].

Statistical analyses were performed with the programming language R (R Core Team, 2019) and the DESeq2 R package [22] Technical replicates where combined and differential gene expression considering the embryonal timepoint was assessed with the DESeq2's Wald test as described in Love et al. [23] We required an alpha level of 0.01 and a minimum log2 fold-change of log2(1.5). Cumulative expression distributions were calculated for *rlog* normalized

expression values for each timepoint separately. We identified the mouse homologous genes of our human genes of interest using the biomaRt R package [24].

## Results

### ES analysis

ES analysis identified 25 apparent *de novo* variants in 25 genes in 18 unrelated case-parent trios. Confirmation of these variants using Sanger sequencing validated all of them and confirmed 23 as being *de novo* in patients. 14 of these variants were novel according to the "Genome Aggregation Database (gnomAD; https://gnomad.broadinstitute.org/; November 2019)". In addition, eight of the confirmed *de novo* variants were found to be rare with a minor allele frequency (MAF) between 0.000003–0.00003 (Table 1). One confirmed *de novo* variant in *TPP2* (c.1534G>A, p.Val512Ile, NM_003291.2, rs73578896) has been previously reported in gnomAD with a MAF of 0.002 (303/275.928) and was therefore filtered out. Through personal communication during the project (Dr. Julia Höfele, Institute of Human Genetics, Klinikum Rechts der Isar, Technical University of Munich, School of Medicine, Munich, Germany) we identified an additional EA/TEF case-parent trio (750_501) in which the patient carries a rare *de novo* variant in *ZFHX3* (c.6377C>T, p.Ala2126Val, allele frequency 0.000019) (marked with an asterisks in Table 1).

**Table 1. Prioritized *de novo* variants.**

| Ext-Code | Phenotype | Variant | HGNC | Refseq | gnomAD (MAF) | MutCDNA | MutProt | Mm | Gg | Dr | Xt |
|---|---|---|---|---|---|---|---|---|---|---|---|
| 4_501 | VATER/VACTERL-like association | 1 | *EEF1D* | NM_032378.4 | | **c.874C>T** | **p.Arg292**\* | K | E | | K |
| | | 2 | *CELSR1* | NM_014246.1 | 3/282,594 (0.00001) | c.4357G>A | p.Val1453Ile | V | I | | I |
| 21_501 | nonsyndromic | 3 | *HPS3* | NM_032383.3 | 10/282,776 (0.00004) | c.1189C>T | p.Arg397Trp | H | R | R | R |
| 27_501 | nonsyndromic | 4 | *PIGC* | NM_153747.1 | | c.716C>T | p.Ala239Val | A | | G | A |
| 35_501 | VATER/VACTERL-like association | 5 | *NFX1* | NM_002504.4 | | c.1723G>A | p.Val575Met | V | | | V |
| 36_501 | nonsyndromic | 6 | ***ZFHX3*** | NM_006885.3 | | c.1601C>G | p.Pro534Arg | P | P | P | N |
| 41_501 | VATER/VACTERL-like association | 7 | *MTA3* | NM_020744.2 | 1/237,600 (0.000004) | c.393C>A | p.Phe131Leu | F | | F | |
| 46_501 | nonsyndromic | 8 | ***FANCB*** | NM_152633.2 | | c.782G>A | p.Arg261Gln | R | Q | S | |
| | | 9 | *PLEC* | NM_201379.1 | 17/272,690 (0.00006) | c.6704G>A | p.Arg2394His | R | R | K | R |
| 63_501 | VATER/VACTERL-like association | 10 | *PPIP5K2* | NM_015216.2 | 2/247,732 (0.000008) | c.686G>A | p.Arg229Gln | **R** | **R** | **R** | **R** |
| 88_501 | nonsyndromic | 11 | *CLP1* | NM_006831.2 | 1/251,486 (0.000003) | c.814C>A | p.His272Asn | **H** | **H** | **H** | **H** |
| | | 12 | *GPR133* | NM_198827.3 | 6/282,534 (0.00002) | c.1033G>A | p.Ala345Thr | A | | | |
| | | 13 | *SLC5A2* | NM_003041.3 | | c.644T>C | p.Leu215Pro | L | | L | L |
| 90_501 | VATER/VACTERL-like association | 14 | ***KIAA0556*** | NM_015202.2 | | c.3730C>T | p.His1244Tyr | **H** | **H** | **H** | **H** |
| 141_501 | VATER/VACTERL-like association | 15 | *STAB1* | NM_015136.2 | 9/278,948 (0.00003) | c.6145C>T | p.Arg2049Cys | R | | S | |
| 154_501 | VATER/VACTERL association | 16 | *GGT6* | NM_153338.2 | | c.1045A>G | p.Ser349Gly | S | | | |
| 167_501 | nonsyndromic | 17 | ***CHD7*** | NM_017780.3 | | c.4187C>G | p.Ala1396Gly | A | A | A | |
| 172_501 | VATER/VACTERL-like association | 18 | *NPR2* | NM_003995.3 | | c.952C>G | p.Arg318Gly | R | K | | T |
| 174_501 | nonsyndromic | 19 | *UBA3* | NM_198195.1 | | c.1088C>T | p.Ser363Phe | S | S | T | P |
| 181_501 | nonsyndromic | 20 | *TANC2* | NM_025185.3 | | c.2357C>T | p.Pro786Leu | P | | P | P |
| 288_501 | VATER/VACTERL association | 21 | ***TRPS1*** | NM_014112.2 | | **c.1630C>T** | **p.Arg544**\* | R | | R | R |
| | | 22 | *APOL2* | NM_145637.1 | | c.319G>C | p.Glu107Gln | D | | | |
| 750_501\* | VATER/VACTERL association | 23 | ***ZFHX3*** | NM_006885.3 | 5/250,880 (0.00002) | c.6377C>T | p.Ala2126Val | A | T | T | A |

Annotations marked in bold red represent: "known disease genes" involved in the formation of congenital malformations, variants with truncating consequence, variants in highly conserved regions of the protein, or novel variants (not found in (n.f.i.), gnomAD (MAF)).

Among the novel variants (i) five reside within previously described disease genes (*CHD7*, *FANCB*, *TRPS1*, *KIAA0556*, and *ZFHX3*), (ii) two variants were truncating (c.874C>T (p. Arg292*) in *EEF1D* and c.1630C>T (p.Arg544*) in *TRPS1*), and (iii) three amino acid changes (p.Arg229Gln in *PPIP5K2*; p.His272Asn in *CLP1*; p.His1244Ytyr in *KIAA0556*) reside in highly conserved regions of the respective protein (Table 1). Of the novel *de novo* variants constituting missense variants four amino acid changes (p.Pro534Arg in *ZFHX3*; p.Phe131Leu in *MTA3*; p.Leu215Pro in *SLC5A2*; p.Ala1396Gly in *CHD7*) were called deleterious by at least seven out of nine *in silico* prediction tools (written in bold in Table 2). Similarly, the rare *de novo* variant in *ZFHX3* (c.6377C>T, p.Ala2126Val, allele frequency 0.000019) found in the additional case-parent trio (750_501), was also called deleterious by seven out of nine *in silico* prediction tools (written in bold in Table 2).

One of the novel *de novo* variants (*PIGC*, c.716C>T, CADD score 11,6) and three of the rare *de novo* variants (*CELSR1*, c.4357G>A, CADD score 15.3; *CLP1*, c.814C>A, CADD score 17.4; *ZFHX3*, c.6377C>T, CADD score 19.2) reached CADD scores between 10 and 20 indicating that these variants have been predicted to be among the 10% most deleterious substitutions within the human genome. Nine of the novel *de novo* variants (*EEF1D*, c.874C>T, CADD score 28.5; *NFX1*, c.1723G>A, CADD score 20.8; *ZFHX3*, c.1601C>G, CADD score 22.3; *SLC5A2*, c.644T>C, CADD score 27.6; *CHD7*, c.4187C>G, CADD score 33; *NPR2*, c.952C>G, *CADD* score 22.2, *UBA3*, c.1088C>T, CADD score 27.8, *TANC2*, c.2357C>T, CADD score 22.7; *TRPS1*, c.1630C>T, CADD score 36) and five of the rare *de novo* variants (*HPS3*, c.1189C>T, CADD score 35; MTA3, c.393C>A, CADD score 26; *PLEC*, c.6704G>A, CADD score 26.5; *PPIP5K2*, c.686G>A, CADD score 34; *STAB1*, c.6145C>T, CADD score 24.1) reached CADD scores over 20 indicating that these variants are predicted to be among the 1% most deleterious variants in the human genome (written in bold in Table 2).

## Re-sequencing of *ZFHX3* in EA/TEF patients

Re-sequencing of *ZFHX3* in 192 EA/TEF patients did not identify additional putative disease-causing variants.

## Structural modeling and in-silico analysis of ZFHX3 protein variants

Swiss model employed template id 3wbj.1.A as a template and built the ZFHX3 amino acid change p.Pro534Arg. Structural models were obtained with sequence identity 14.89%, coverage of 58.75%, and normalized Z-score of -2.90. The respective values are considered as an indicative of correctly folded and good modeled structures close to native structure. For the amino acid change p.Ala2126Val, structural models were obtained with sequence identity of 19.15%, coverage 27.48%, and normalized Z-score of -1.76. From the structural modeling of the ZFHX3 amino acid changes, we found that the two changes do not have any distortion in the native protein amounting to RMSD change at α-carbon is 0.02 Å and at backbone is 0.03 Å in p.Pro534Arg and RMSD change of at α-carbon is 0.05 Å and at backbone is 0.06 Å in p. AlaA2126Val (S1–S4 Figs).

## Transcriptome analysis

Evaluation of the transcriptome data showed that all murine genes were expressed at E8.5, E12.5, and postnatal except for some *APOL2* orthologous. Differential gene expression analysis revealed that four out of 24 genes were transcriptome-wide differentially expressed between the time points of E8.5 and E12.5 (*Chd7*: logFC 2.243, p.adj 1.85E-29; *Npr2*: logFC -2.268, p. adj 5.18E-04; *Trps1*: logFC -2.927, p.adj 1.17E-28; *Eef1d*: logFC 1.248, p.adj 6.41E-05), and two between the time points E12.5 and postnatal (*Apol7a*: logFC -9.273, p.adj 5.18E-07; *Plec*: logFC

**Table 2. Classification of *de novo* variants using in silico prediction programs.**

| Ext-Code | Variant | HGNC | MutCDNA | gnomAD (MAF) | SIFT | LRT | Mutation Taster | Mutation Assessor | FATHMM | PROVEAN | Meta SVM | Meta LR | Fathmm MKL_coding | CADD Score |
|---|---|---|---|---|---|---|---|---|---|---|---|---|---|---|
| 4_501 | 1 | *EEF1D* | c.874C>T | | - | N | A | - | - | - | - | - | N | **28,5** |
| | 2 | *CELSR1* | c.4357G>A | 3/282,594 (0.00001) | T | U | N | N | T | N | T | T | D | 15,3 |
| 21_501 | 3 | *HPS3* | c.1189C>T | 10/282,776 (0.00004) | D | D | A | M | T | D | D | T | D | **35** |
| 27_501 | 4 | *PIGC* | c.716C>T | | T | D | D | M | T | N | T | T | D | 11,6 |
| 35_501 | 5 | *NFX1* | c.1723G>A | | D | N | N | L | T | N | T | T | D | **20,8** |
| 36_501 | 6 | *ZFHX3* | c.1601C>G | | D | N | D | L | T | N | T | T | D | **22,3** |
| 41_501 | 7 | *MTA3* | c.393C>A | 1/237,600 (0.000004) | **D** | **D** | **D** | H | **D** | **D** | **D** | **D** | **D** | **26** |
| 46_501 | 8 | *FANCB* | c.782G>A | | T | N | N | N | T | N | T | T | N | 7,2 |
| | 9 | *PLEC* | c.6704G>A | 17/272,690 (0.00006) | D | U | D | N | T | N | T | T | D | **26,5** |
| 63_501 | 10 | *PPIP5K2* | c.686G>A | 2/247,732 (0.000008) | D | D | D | M | T | D | T | T | D | **34** |
| 88_501 | 11 | *CLP1* | c.814C>A | 1/251,486 (0.000003) | T | D | D | L | T | N | T | T | D | 17,4 |
| | 12 | *GPR133* | c.1033G>A | 6/282,534 (0.00002) | T | N | N | N | T | N | T | T | N | 0,016 |
| | 13 | *SLC5A2* | c.644T>C | | **D** | **D** | **D** | H | **D** | **D** | **D** | **D** | **D** | **27,6** |
| 90_501 | 14 | *KIAA0556* | c.3730C>T | | D | N | N | L | T | N | T | T | N | 1,9 |
| 141_501 | 15 | *STAB1* | c.6145C>T | 9/278,948 (0.00003) | T | N | N | M | T | D | T | T | N | **24,1** |
| 154_501 | 16 | *GGT6* | c.1045A>G | | D | N | N | N | T | D | T | T | N | 5,9 |
| 167_501 | 17 | *CHD7* | c.4187C>G | | **D** | **D** | **D** | H | T | **D** | **D** | **D** | **D** | **33** |
| 172_501 | 18 | *NPR2* | c.952C>G | | T | N | D | L | D | D | T | T | D | **22,2** |
| 174_501 | 19 | *UBA3* | c.1088C>T | | D | D | D | L | T | D | T | T | D | **27,8** |
| 181_501 | 20 | *TANC2* | c.2357C>T | | T | D | D | L | T | D | T | T | D | **22,7** |
| 288_501 | 21 | *TRPS1* | c.1630C>T | | - | D | A | - | - | - | - | - | D | **36** |
| | 22 | *APOL2* | c.319G>C | | T | N | N | N | T | N | T | T | N | 0,004 |
| 750_501* | 23 | *ZFHX3* | c.6377C>T | 5/250,880 (0.00002) | **D** | **D** | **D** | L | T | **D** | **D** | **D** | **D** | 19,2 |

*A: automatic disease causing; D: disease causing; H: high functional; L: non-functional; M: medium functional; N: neutral; T: tolerant. Annotations marked in bold red represent: "variants that are classified to be disease causing by at least eight out of ten *in silico* prediction programs (except for truncating variants) used by dbNSFP v3.0 (https://sites.google.com/site/jpopgen/dbNSFP)".

-2.352, p.adj 9.09E-24). Interestingly, most of the candidate genes were highly expressed at each time point (Fig 1, Fig 2, S2 Table). The candidate genes *Zfhx3*, *Ppip5k2*, *Chd7* and *Eef1d* were even expressed above the 95[th] percentile at E8.5 compared to the expression of all other genes. In addition, the genes *Ppip5k2*, *Trps1*, *Zfhx3* and *Eef1d* were expressed above the 93[rd] percentile at E12.5.

## Discussion

The etiology of EA/TEF is heterogeneous. Previously, disease causing monoallelic mutations of variable genomic size have been reported among EA/TEF patients [7; 25] Here, we identified 23 single nucleotide *de novo* variants in 23 different genes using 30 unrelated case-parent trios and ES. All confirmed *de novo* variants were prioritized using *in silico* prediction tools.

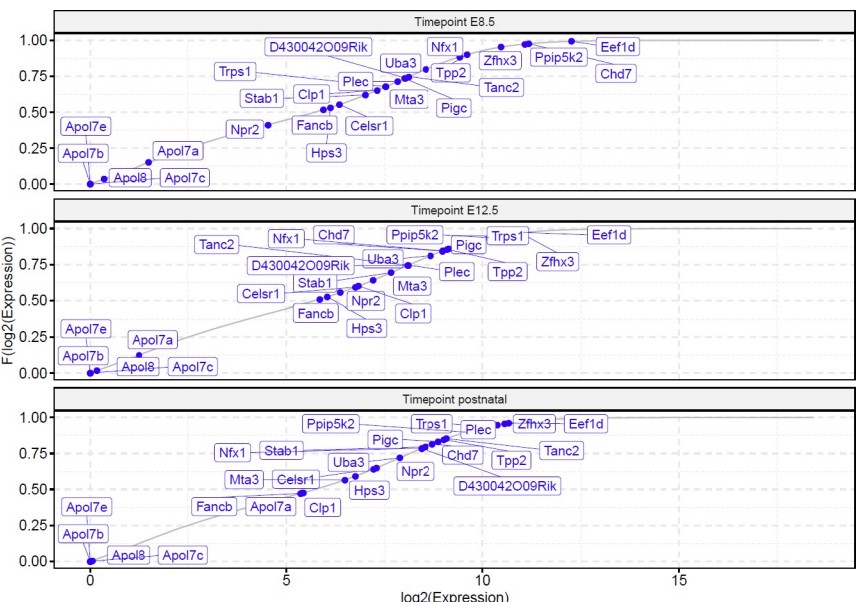

**Fig 1. Empirical cumulative distribution of murine candidate gene expression at each timepoint.** The empirical cumulative distribution function (F) was calculated from Regularized Log (*rlog*) transformed expression values. E: embryonic day.

The embryonic role of genes harboring prioritized *de novo* variants was further investigated by targeted analysis of mouse transcriptome data of esophageal tissue obtained at E8.5, E12.5, and postnatal.

After prioritization of variants using *in silico* prediction tools, targeted analysis of mouse transcriptome data, and review of the literature we prioritize *TRPS1* and *ZFHX3* as new EA/

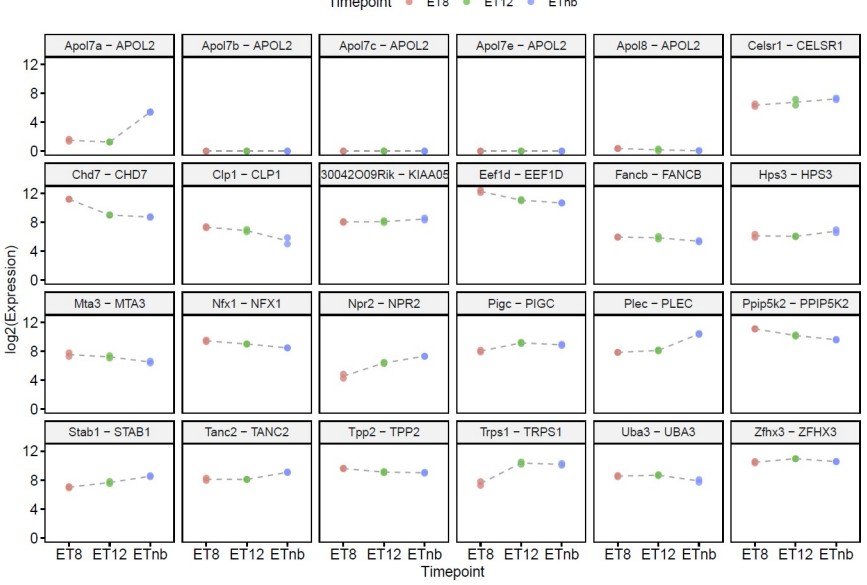

**Fig 2. Murine candidate gene expression at three different timepoints.** The gene expression is shown as log2 expression on the y-axis, while the timepoint is shown as categorial variable on the x-axis. The header of each sub figure shows the murine and human gene symbols. E: embryonic day, pn: postnatal.

TEF candidate genes and provide further support for *CHD7* as a key player in esophageal development.

The identified *de novo* amino acid change in *CHD7* has not been previously described. *CHD7* has been established as the major disease gene for CHARGE syndrome (OMIM #214800) [26]. Eight out of nine *in silico* prediction programs used by dbNSFP v3.0 (https://sites.google.com/site/jpopgen/dbNSFP) classified this *de novo* amino acid change p. Ala1396Gly in *CHD7* as deleterious. As about 20% of patients with CHARGE syndrome present with EA/TEF [27] we consider the identified variant as disease causing in our patient (167_501, Table 1) even though patient 167_501 did not present with additional congenital anomalies besides EA/TEF that would have suggested the clinical diagnosis of CHARGE syndrome. None of the other identified *de novo* variants resided within a gene that was previously linked to the formation of EA/TEF.

In patient 288_501 we identified a novel *de novo* truncating amino acid change p.Arg544* in *TRPS1* associated with tricho-rhino-phalangeal syndrome I (OMIM #190350). Previously Maas et al. (2015) reported the same truncating variant in three unrelated patients with TRPS1 [25]. Unlike our patient 288_501, these previously reported patients did not present with any congenital anomaly of esophagus or trachea nor with any congenital anomaly of the heart (personal communication with Dr. Raoul C. Hennekam). Interestingly, in the here generated expression data *Trps1* shows a consistently high expression levels of 67th percentile at E8.5 and a log2 Foldchange of -2.92 between days E8.5 and E12.5 in mouse embryos in the esophageal area, suggestive of an involvement of Trps1 during vertebrate foregut development. The latter hypothesis suggests that the here identified *de novo* variant in *TRPS1* might be involved in the expression of EA/TEF in patient 288_501.

In patient 36_501 with nonsyndromic EA/TEF we identified a novel *de novo* variant in *ZFHX3*. Through personal communication during the project we identified another patient with EA/TEF as part of his VATER/VACTERL association (patient 750_501) with a *de novo* variant in *ZFHX3*. While the novel variant p.Pro534Arg resides in a well-conserved region of *ZFHX3* and has not been reported in gnomAD, the variant p.Ala2126Val has been reported five times heterozygous in 250,880 alleles in gnomAD (MAF 0.00002) (Table 1) and resides in a less well conserved region of *ZFHX3*. Prompted by this finding, we re-sequenced *ZFHX3* in 192 additional EA/TEF patients but did not find any further putative EA/TEF associated variant. In order to further analyze the two identified amino acid changes in ZFHX3 we further performed structural modeling and *in-silico* analysis of ZFHX3 protein. Here, substitution of C to G at position c.1601 has resulted in substitution of Pro to Arg at position 534 with a RMSD value amounting to 0.02Å at C-alpha carbon and 0.03Å in the protein backbone. Similarly, for the C to T substitution at c.6377 position that resulted into Ala to Val substitution at position 2126 has also recorded a similar change in RMSD value, 0.05Å at C-alpha carbon and 0.06Å in the protein backbone. While the structural modeling suggests that both changes do not cause distortion of the native protein, a possible functional impact of both variants would warrant further functional testing. According to our transcriptome analysis *Zfhx3* is not differentially expressed between either E8.5 and E12.5 or E12.5 and postnatal. However, *Zfhx3* is among the top expressed genes at E8.5 (>95th percentile) and E12.5 (>97rd percentile). Interestingly, Thisse and Thisse (2004) reported also expression of *zfhx3* in zebrafish larvae 24 hours post fertilization in the region of the pharyngeal arches representing a series of paired bony or cartilaginous arches that develop along the lateral walls of the foregut, supporting the role of *ZFHX3* in vertebrate foregut development [28]. Taken together, the here detected *de novo* variants in human EA/TEF patients, the high *Zfhx3* expression at 8.5 and 12.5 in embryonic foregut tissue of mouse embryos and the previously reported expression of *zfhx3* in

zebrafish larvae in the region of the pharyngeal arches suggests *ZFHX3* as a putative EA/TEF candidate gene.

Overall, interpretation of the data is limited by the lack of animal models, at least for the findings in *CHD7*, *TRPS1*, and *ZFHX3*. To the best of our knowledge, there has no animal model been described that would have investigated embryonic foregut development, when these genes have been deleted. In order to definitely conclude that our findings respectively *de novo* variants in *CHD7*, *TRPS1*, and *ZFHX3* have been directly causative for the EA/TEF phenotype in the respective patients, *in vivo* experiments including animal models would be necessary, which were beyond the scope of our present study.

## Conclusion

In summary, we detected 23 *de novo* mutations in 23 genes in 17 unrelated patients. Human exome and mouse embryonic expression analyses suggest *ZHFX3* and *TRPS1* as putative EA/TEF candidate genes and endorse *CHD7* as a key player for esophageal development.

## Supporting information

**S1 Fig. Wild and mutant structures of ZFHX3 (c.1601C>G).**
(PNG)

**S2 Fig. Super imposed structure of ZFHX3 Wild & mutant (c.1601C>G).**
(PNG)

**S3 Fig. Wild and mutant structures of ZFHX3 (c.6377C>T).**
(PNG)

**S4 Fig. Super imposed structure of ZFHX3 Wild & mutant (c.6377C>T).**
(PNG)

**S1 Table. Phenotypes of the patients included in the ES.** VATER/VACTERL-like association (vertebral defects (V), anorectal malformations (A), cardiac defects (C), tracheoesophageal fistula with or without esophageal atresia (TE), renal malformations (R), and limb defects (L) [5], Ventricular septal defect (VSD), atrial septal defect (ASD).
(DOCX)

**S2 Table. logFC: log2 of foldchange, baseMean: Average gene expression across all timepoints, pvalue: nominal P-value, padj: Benjamini-Hochberg corrected P-value, ECDF percentile: Percentile of empirical cumulative distribution function for each timepoint.**
(DOCX)

**S1 Data.**
(DOCX)

## Acknowledgments

We thank all patients and their families for their participation, as well as the German self-help organizations for individuals with anorectal malformations (SoMA e.V.) and tracheoesophageal fistula with or without esophageal atresia (TE) (KEKS e.V.) for their assistance with recruitment. We also thank Prof. Raoul Hennekam for fruitful discussion on the manuscript.

## Author Contributions

**Conceptualization:** Jan Gehlen, Tikam Chand Dakal, Alice Hölscher, Phillip Grote, Holger Thiele, Heiko Reutter.

**Data curation:** Jan Gehlen, Amit Kawalia, Alice Hölscher, Stefanie Heilmann-Heimbach, Nadine Zwink, Ekkehart Jenetzky, Phillip Grote, Holger Thiele, Heiko Reutter.

**Formal analysis:** Rong Zhang, Jan Gehlen, Amit Kawalia, Maria-Theodora Melissari, Tikam Chand Dakal, Athira M. Menon, Julia Höfele, Korbinian Riedhammer, Lea Waffenschmidt, Amit Sharma, Michael Ludwig, Holger Thiele, Heiko Reutter.

**Funding acquisition:** Julia Höfele, Ekkehart Jenetzky, Phillip Grote, Johannes Schumacher, Holger Thiele, Heiko Reutter.

**Investigation:** Rong Zhang, Maria-Theodora Melissari, Tikam Chand Dakal, Athira M. Menon, Lea Waffenschmidt, Julia Fabian, Katinka Breuer, Jeshurun Kalanithy, Alina Christine Hilger, Amit Sharma, Michael Ludwig.

**Methodology:** Jan Gehlen, Tikam Chand Dakal, Holger Thiele.

**Project administration:** Nadine Zwink, Ekkehart Jenetzky, Phillip Grote, Holger Thiele, Heiko Reutter.

**Resources:** Jeshurun Kalanithy, Alice Hölscher, Thomas M. Boemers, Markus Pauly, Andreas Leutner, Jörg Fuchs, Guido Seitz, Barbara M. Ludwikowski, Barbara Gomez, Jochen Hubertus, Andreas Heydweiller, Ralf Kurz, Johannes Leonhardt, Ferdinand Kosch, Stefan Holland-Cunz, Oliver Münsterer, Beno Ure, Eberhard Schmiedeke, Jörg Neser, Petra Degenhardt, Stefanie Märzheuser, Katharina Kleine, Mattias Schäfer, Nicole Spychalski, Oliver J. Deffaa, Jan-Hendrik Gosemann, Martin Lacher, Stefanie Heilmann-Heimbach, Nadine Zwink, Ekkehart Jenetzky, Phillip Grote, Johannes Schumacher, Holger Thiele, Heiko Reutter.

**Software:** Holger Thiele.

**Supervision:** Michael Ludwig, Phillip Grote, Holger Thiele, Heiko Reutter.

**Validation:** Rong Zhang, Phillip Grote.

**Visualization:** Jan Gehlen, Holger Thiele, Heiko Reutter.

**Writing – original draft:** Rong Zhang, Jan Gehlen, Tikam Chand Dakal, Julia Höfele, Amit Sharma, Michael Ludwig, Phillip Grote, Johannes Schumacher, Holger Thiele, Heiko Reutter.

**Writing – review & editing:** Rong Zhang, Jan Gehlen, Tikam Chand Dakal, Korbinian Riedhammer, Alina Christine Hilger, Amit Sharma, Markus Pauly, Andreas Leutner, Jörg Fuchs, Guido Seitz, Barbara M. Ludwikowski, Barbara Gomez, Jochen Hubertus, Nadine Zwink, Ekkehart Jenetzky, Phillip Grote, Johannes Schumacher, Heiko Reutter.

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
