## [Decision Letter · Decision Letter 0]

26 Mar 2020

PONE-D-20-03485

Human exome and mouse embryonic expression data implicate ZFHX3, TRPS1, and CHD7 in human esophageal atresia

PLOS ONE

Dear Prof. Reutter,

Thank you for submitting your manuscript to PLOS ONE. After careful consideration, we feel that it has merit but does not fully meet PLOS ONE’s publication criteria as it currently stands. Therefore, we invite you to submit a revised version of the manuscript that addresses the points raised during the review process.

We would appreciate receiving your revised manuscript by May 10 2020 11:59PM. To enhance the reproducibility of your results, we recommend that if applicable you deposit your laboratory protocols in protocols.io, where a protocol can be assigned its own identifier (DOI) such that it can be cited independently in the future. For instructions see: http://journals.plos.org/plosone/s/submission-guidelines#loc-laboratory-protocols

We look forward to receiving your revised manuscript.

Kind regards,

Regie Lyn Pastor Santos-Cortez, M.D., Ph.D.

Academic Editor

PLOS ONE

Journal Requirements:

2. Please include your tables as part of your main manuscript and remove the individual files. Please note that supplementary tables (should remain/ be uploaded) as separate "supporting information" files

3. Please note that PLOS does not permit references to “data not shown.” Authors should provide the relevant data within the manuscript, the Supporting Information files, or in a public repository. If the data are not a core part of the research study being presented, we ask that authors remove any references to these data.

Reviewers' comments:

Reviewer's Responses to Questions

**Comments to the Author**

1. Is the manuscript technically sound, and do the data support the conclusions?

Reviewer #1: Yes

Reviewer #2: Yes

Reviewer #3: Yes

2. Has the statistical analysis been performed appropriately and rigorously? 

Reviewer #1: Yes

Reviewer #2: Yes

Reviewer #3: Yes

3. Have the authors made all data underlying the findings in their manuscript fully available?

Reviewer #1: Yes

Reviewer #2: Yes

Reviewer #3: Yes

4. Is the manuscript presented in an intelligible fashion and written in standard English?

Reviewer #1: Yes

Reviewer #2: Yes

Reviewer #3: Yes

5. Review Comments to the Author

Reviewer #1: The present study examined the exome of EA/TEF patients and parents (30 trios) aiming at identifying mutational de novo events and the variant forms were prioritized using prediction tools. Then, the embryonic role of the genes with prioritized de novo variants was examined in oesophageal tissue of mice obtained at E8.5, E12.5 and after birth by targeted analysis of the transcriptome.

A total of 14 novel de novo variants in 14 genes and 8 rare de novo variants in another 8 genes were identified. After comparative analysis of the transcriptome of the mouse tissue the authors conclude that CHD7, TRPS1, and ZFHX3 are EA/TEF candidate genes. The conclusion is that rare mutational de novo events involved in foregut embryogenesis contribute to the development of EA/TEF.

The study is well planned and well executed. The collaborative effort to obtain material from a large number of patient/parent trios of this relatively rare disease from several locations is to be commended and the idea of contrasting the findings with mice tissue is an interesting approach.

The introduction is good. M&M are clearly described. The results are well displayed in text and tables and figures. The discussion is OK and the conclusions are sound. The references, figures and tables are OK.

I think that this material deserves publication but, prior to it, I would like to have some additional information and suggest a minor change in the text.

1. Normal mouse material from E8.5 to E12.5 involves the entire tracheoesophageal cleavage period during which, in case of certain disturbances, EA/TEF is generated. However, at E8.5 there is no oesophagus as such but rather a common foregut tube. In contrast, at day 12.5 the oesophagus is completely separated. It would be useful to have some more details of how microdissection of the material of the foregut and surrounding mesenchyme was made and what part was investigated. The interaction of several genes and transcription factors at epithelial and mesenchymal levels play a role in tracheoesophageal separation. Did samples on E8.5 include the surrounding mesenchyme?. Did samples on E12.5 include the trachea?. EA/TEF is a digestive and respiratory malformation and omitting the respiratory material could lead to incorrect interpretation. Some more details would be appreciated.

2. According to the authors, the mice were sacrificed by cervical "translocation" (to put something at another location). The right word, I believe is "dislocation".

Reviewer #2: The manuscript titled “Human exome and mouse embryonic expression data implicate ZFHX3, TRPS1, and CHD7in human esophageal atresia” is a very interesting study of new insight into pathogenesis of esophageal atresia in human.

Esophageal atresia (OA) and tracheoesophageal fistula (TOF) are relatively frequently occurring foregut malformations whose etiology and pathogenesis are heterogeneous and not clearly understood. Advances in surgical techniques and perioperative care have increased survival rates to over 95% for isolated cases.

The pathogenesis of esophageal atresia is heterogeneous. It is thought that a combination of genetic and environmental factors play a role in the etiology of foregut anomalies and it is most likely. However, recent results from molecular genetic studies on esophageal atresia have yielded a greater understanding of the molecular mechanism involved foregut morphogenesis.

Study of OA patients using trio exome sequencing (patient and parents) associated with the study with mouse transcriptome data of esophageal tissue, are very important step in identifying and understanding etiological factor for this congenital defect.

The study have very well scientific plan and was well performed. The abstract, background and methods sections are clearly written. The result and discussion section is also are very clearly presented.

I have neither major nor minor comments.

Reviewer #3: This is an interesting paper which aims to find out genes involved in the development of esophageal atresia tracheo esophageal fistula.

The authors recruited 30 trios and performed exome sequencing, in silico testing of the function of proteins resulting from their genetic analysis and tried to validate their finding by looking at differential expression of genes in mice anterior foregut endoderm at E8.5 (just before separation into esophagus and trachea) and at E12.5 (after separation).

Although interesting, the study, essentially descriptive, does not bring significant new information except hypothetic genes which could be involved in EA/TEF. I would suggest to use animal models (zebrafish?) to bring mechanistic data to definitely show the roles of those genes.

Specific questions:

1- How were recruited the 30 trios?

2-How many mouse embryos were pooled for RNA seq? What does mean 'two biological samples were obtained for each time point'?

6. PLOS authors have the option to publish the peer review history of their article (what does this mean?). If published, this will include your full peer review and any attached files.

Reviewer #1: No

Reviewer #2: Yes: Robert Smigiel

Reviewer #3: No

---

## [Author Response · Author response to Decision Letter 0]

10 Apr 2020

PONE-D-20-03485

We thank the reviewers for their constructive comments. Following the comments, we have addressed all points and our point-by-point responses are provided below. In each case, we indicate where and how the manuscript has been amended. In the revised manuscript, all changes to the original text are written in red. Previous phrasings have been crossed out. We hope that our manuscript will now be considered suitable for publication.

Reviewer #1: 

The present study examined the exome of EA/TEF patients and parents (30 trios) aiming at identifying mutational de novo events and the variant forms were prioritized using prediction tools. Then, the embryonic role of the genes with prioritized de novo variants was examined in oesophageal tissue of mice obtained at E8.5, E12.5 and after birth by targeted analysis of the transcriptome. A total of 14 novel de novo variants in 14 genes and 8 rare de novo variants in another 8 genes were identified. After comparative analysis of the transcriptome of the mouse tissue the authors conclude that CHD7, TRPS1, and ZFHX3 are EA/TEF candidate genes. The conclusion is that rare mutational de novo events involved in foregut embryogenesis contribute to the development of EA/TEF. The study is well planned and well executed. The collaborative effort to obtain material from a large number of patient/parent trios of this relatively rare disease from several locations is to be commended and the idea of contrasting the findings with mice tissue is an interesting approach. The introduction is good. M&M are clearly described. The results are well displayed in text and tables and figures. The discussion is OK and the conclusions are sound. The references, figures and tables are OK. I think that this material deserves publication but, prior to it, I would like to have some additional information and suggest a minor change in the text.

Comment 1: 

Normal mouse material from E8.5 to E12.5 involves the entire tracheoesophageal cleavage period during which, in case of certain disturbances, EA/TEF is generated. However, at E8.5 there is no oesophagus as such but rather a common foregut tube. In contrast, at day 12.5 the oesophagus is completely separated. It would be useful to have some more details of how microdissection of the material of the foregut and surrounding mesenchyme was made and what part was investigated. The interaction of several genes and transcription factors at epithelial and mesenchymal levels play a role in tracheoesophageal separation. Did samples on E8.5 include the surrounding mesenchyme?. Did samples on E12.5 include the trachea?. EA/TEF is a digestive and respiratory malformation and omitting the respiratory material could lead to incorrect interpretation. Some more details would be appreciated.

Answer to Comment 1:

Wie thank the reviewer for this comment. We looked carefully again at the section of the manuscript where we provide the information reviewer #1 is referring to. In the materials and methods section, we state for the E8.5 timepoint: “the pharyngeal pouch containing endoderm and adjacent mesoderm tissue was surgically isolated”. We think this explains very well what part we dissected. For the E12.5 and neonatal timepoint we state: “the distinct structure of the esophagus was surgically isolated and transferred into QIAzol®”.

The latter contained an error, which we corrected, and it now reads: “…the distinct structures of the esophagus and the trachea was surgically isolated, combined and transferred into QIAzol®”.

Comment 2: 

According to the authors, the mice were sacrificed by cervical "translocation" (to put something at another location). The right word, I believe is "dislocation".

Answer to Comment 2:

We apologize for this autocorrect issue and now corrected the word “translocation” with “dislocation”.

Reviewer #2: 

The manuscript titled “Human exome and mouse embryonic expression data implicate ZFHX3, TRPS1, and CHD7in human esophageal atresia” is a very interesting study of new insight into pathogenesis of esophageal atresia in human. Esophageal atresia (OA) and tracheoesophageal fistula (TOF) are relatively frequently occurring foregut malformations whose etiology and pathogenesis are heterogeneous and not clearly understood. Advances in surgical techniques and perioperative care have increased survival rates to over 95% for isolated cases. The pathogenesis of esophageal atresia is heterogeneous. It is thought that a combination of genetic and environmental factors play a role in the etiology of foregut anomalies and it is most likely. However, recent results from molecular genetic studies on esophageal atresia have yielded a greater understanding of the molecular mechanism involved foregut morphogenesis. Study of OA patients using trio exome sequencing (patient and parents) associated with the study with mouse transcriptome data of esophageal tissue, are very important step in identifying and understanding etiological factor for this congenital defect. The study have very well scientific plan and was well performed. The abstract, background and methods sections are clearly written. The result and discussion section is also are very clearly presented. I have neither major nor minor comments.

Comment to the Reviewer:

We thank the reviewer for his decent commentary on our study.

Reviewer #3: 

This is an interesting paper which aims to find out genes involved in the development of esophageal atresia tracheo esophageal fistula. The authors recruited 30 trios and performed exome sequencing, in silico testing of the function of proteins resulting from their genetic analysis and tried to validate their finding by looking at differential expression of genes in mice anterior foregut endoderm at E8.5 (just before separation into esophagus and trachea) and at E12.5 (after separation). Although interesting, the study, essentially descriptive, does not bring significant new information except hypothetic genes which could be involved in EA/TEF. I would suggest to use animal models (zebrafish?) to bring mechanistic data to definitely show the roles of those genes. Specific questions:

Comment 1: 

 How were recruited the 30 trios?

Answer to Comment 1:

As requested by the reviewer, we have added additional information to how the 30 case-parent trios were recruited. It now reads: “In 2011, the authors JS and HR founded the scientific network “great” (genetic risk for esophageal atresia; www.great-konsortium.de). The “great network” was founded in order to initiate a nationwide investigation into the genetic causes of EA/TEF. Prior to the commencement of recruitment, the network partners generated a unique standardized case report form (CRF). The CRF comprises an epidemiological questionnaire and a clinical assessment battery. The epidemiological questionnaire is based on: (i) the National Birth Defect Prevention Study questionnaire of the U.S. Centers of Disease Control and Prevention (www.nbdpn.org); and (ii) the questionnaire of the European Surveillance of Congenital Malformations (EUROCAT) network (www.eurocat-network.eu). The clinical assessment battery comprises the classification system of the EA/TEF phenotype according to Gross (1953), and the ICD10 coding with the British Pediatric Association one digit extension (www.eurocat-network.eu/content/EUROCAT-Guide-1.3.pdf) for classification of additional congenital anomalies. The great cohort is being recruited with the support of pediatric surgical departments across Germany, and the German self-help organization for patients and families with EA/TEF (KEKS e.V.; www.keks.org). KEKS e.V. is the largest self-help organization for EA/TEF families in Europe, and supports both the ongoing great investigations and the present proposal. 

The here described study fulfilled the requirement of the Declaration of Helsinki and ethical approval was obtained from the local ethic committee of the Medical Faculty of Bonn (Lfd. Nr. 073/12). Every participating family provided written informed consent. The 30 here reported case-parent trios as well as the EA/TEF cohort for resequencing of ZFHX3, were recruited through the efforts of the scientific network “great”. In 14 of the 30 case-parent trios, EA/TEF occurred isolated/nonsyndromic. In the remaining case-parent trios EA/TEF co-occurred with additional phenotypic features (syndromic cases) mostly belonging to the VATER/VACTERL spectrum (S1 Tbl.). From each case-parent trio, EDTA blood samples were obtained. Genomic DNA was isolated using the Chemagic DNA Blood Kit special (Chemagen, Baesweiler, Germany). Through personal communication we identified another patient with EA/TEF as part of his VATER/VACTERL association (patient 750_501, see Table 1, marked with asterisk).”

Comment 2: 

How many mouse embryos were pooled for RNA seq? What does mean 'two biological samples were obtained for each time point'?

Answer to Comment 2:

We additional information with the following sentence, it now reads: “For the E8.5 stage we pooled biopsies from 5 embryos to prepare the RNA and for the E12.5 and neonates we pooled two each for RNA preparation.”

---

## [Decision Letter · Decision Letter 1]

23 Apr 2020

PONE-D-20-03485R1

Human exome and mouse embryonic expression data implicate ZFHX3, TRPS1, and CHD7 in human esophageal atresia

PLOS ONE

Dear Prof. Reutter,

Thank you for submitting your manuscript to PLOS ONE. After careful consideration, we feel that it has merit but does not fully meet PLOS ONE’s publication criteria as it currently stands. Therefore, we invite you to submit a revised version of the manuscript that addresses the points raised during the review process.

To potentially satisfy the concern of Reviewer 3, please include one or two sentences in the Discussion citing the lack of animal models as well as functional experiments specific to the identified candidate genes or variants as a limitation of the study.

We would appreciate receiving your revised manuscript by Jun 07 2020 11:59PM. To enhance the reproducibility of your results, we recommend that if applicable you deposit your laboratory protocols in protocols.io, where a protocol can be assigned its own identifier (DOI) such that it can be cited independently in the future. For instructions see: http://journals.plos.org/plosone/s/submission-guidelines#loc-laboratory-protocols

We look forward to receiving your revised manuscript.

Kind regards,

Regie Lyn Pastor Santos-Cortez, M.D., Ph.D.

Academic Editor

PLOS ONE

Reviewers' comments:

Reviewer's Responses to Questions

**Comments to the Author**

1. If the authors have adequately addressed your comments raised in a previous round of review and you feel that this manuscript is now acceptable for publication, you may indicate that here to bypass the “Comments to the Author” section, enter your conflict of interest statement in the “Confidential to Editor” section, and submit your "Accept" recommendation.

Reviewer #1: All comments have been addressed

Reviewer #2: All comments have been addressed

Reviewer #3: (No Response)

2. Is the manuscript technically sound, and do the data support the conclusions?

Reviewer #1: (No Response)

Reviewer #2: Yes

Reviewer #3: Yes

3. Has the statistical analysis been performed appropriately and rigorously? 

Reviewer #1: (No Response)

Reviewer #2: Yes

Reviewer #3: I Don't Know

4. Have the authors made all data underlying the findings in their manuscript fully available?

Reviewer #1: (No Response)

Reviewer #2: Yes

Reviewer #3: Yes

5. Is the manuscript presented in an intelligible fashion and written in standard English?

Reviewer #1: (No Response)

Reviewer #2: Yes

Reviewer #3: Yes

6. Review Comments to the Author

Reviewer #1: (No Response)

Reviewer #2: I have no more comments. The manuscript titled “Human exome and mouse embryonic expression data implicate ZFHX3, TRPS1, and CHD7in human esophageal atresia” is worthy to publication in this version.

Reviewer #3: Thanks to the authors for their responses.

Again, the paper would have been stronger with a mechanistic demonstration of the roles of the genes reported as involved in the development of the foregut.

7. PLOS authors have the option to publish the peer review history of their article (what does this mean?). If published, this will include your full peer review and any attached files.

Reviewer #1: No

Reviewer #2: Yes: Robert Smigiel

Reviewer #3: No

---

## [Author Response · Author response to Decision Letter 1]

12 May 2020

PONE-D-20-03485R1

We thank the reviewer for his comments. We have addressed this comment below. In the revised manuscript, all changes to the earlier revised version are written in red. We hope that our manuscript will now be considered suitable for publication.

Reviewer #3: 

Comment: 

Again, the paper would have been stronger with a mechanistic demonstration of the roles of the genes reported as involved in the development of the foregut.

Answer to Comment 1:

We agree with the reviewer that any molecular genetic finding detected in human individuals with congenital malformations should in the following workup of these findings warrant in vivo functional studies, e.g. animal models to provide further evidence for the involvement of the identified molecular genetic findings in the expression of the disease. However, the creation of animal models was beyond the scope of the present study. In order to provide some level of functional evidence, we generated mouse transcriptome data from wildtype mice, in order to see, if the identified potential disease genes are actually expressed during the embryonic critical time frame. Here we were able to show, that Zfhx3 is continuously high expressed during all time points, and that CHD7 and Trps1 are differentially expressed over all three time points. 

To address the concern of the reviewer, we have added the following sentence to the end of the discussion: “Overall, interpretation of the data is limited by the lack of animal models, at least for the findings in CHD7, TRPS1, and ZFHX3. To the best of our knowledge, there has no animal model been described that would have investigated embryonic foregut development, when these genes have been deleted. In order to definitely conclude that our findings respectively de novo variants in CHD7, TRPS1, and ZFHX3 have been directly causative for the EA/TEF phenotype in the respective patients, in vivo experiments including animal models would be necessary, which were beyond the scope of our present study.”

---

## [Editor Report · Decision Letter 2]

22 May 2020

Human exome and mouse embryonic expression data implicate ZFHX3, TRPS1, and CHD7 in human esophageal atresia

PONE-D-20-03485R2

Dear Dr. Reutter,

We are pleased to inform you that your manuscript has been judged scientifically suitable for publication and will be formally accepted for publication once it complies with all outstanding technical requirements.

With kind regards,

Regie Lyn Pastor Santos-Cortez, M.D., Ph.D.

Academic Editor

PLOS ONE
---

## [Editor Report · Acceptance letter]

28 May 2020

PONE-D-20-03485R2 

Human exome and mouse embryonic expression data implicate *ZFHX3*, *TRPS1*, and *CHD7* in human esophageal atresia 

Dear Dr. Reutter:

I am pleased to inform you that your manuscript has been deemed suitable for publication in PLOS ONE. Congratulations! Your manuscript is now with our production department. 

With kind regards,

on behalf of

Dr. Regie Lyn Pastor Santos-Cortez 

Academic Editor

PLOS ONE